# Beyond identity: Understanding the contribution of the 5' nucleotide of the antisense strand to RNAi activity

Peizhen Yang[1][*], Ericka Havecker[2], Matthew Bauer[2], Carl Diehl[1], Bill Hendrix[3][¤a], Paul Hoffer[3][¤b], Timothy Boyle[2], John Bradley[2], Amy Caruano-Yzermans[2], Jill Deikman[3]

**1** Bayer Crop Science, St. Louis, Missouri, United States of America, **2** Bayer Crop Science, Chesterfield, Missouri, United States of America, **3** Bayer Crop Science, Woodland, California, United States of America

☯ These authors contributed equally to this work.
¤a Current address: Bayer Crop Science, West Sacramento, California, United States of America
¤b Current address: California Governor's Office of Emergency Services, Mather, California, United States of America
* peizhen.yang@bayer.com

**Data Availability Statement:** All raw image files are available from the Zenodo database (https://doi.org/10.5281/zenodo.5227677). All other relevant

## Abstract

In both the pharmaceutical and agricultural fields, RNA-based products have capitalized upon the mechanism of RNA interference for targeted reduction of gene expression to improve phenotypes and traits. Reduction in gene expression by RNAi is the result of a small interfering RNA (siRNA) molecule binding to an ARGONAUTE (AGO) protein and directing the effector complex to a homologous region of a target gene's mRNA. siRNAs properties that govern RNA-AGO association have been studied in detail. The siRNA 5' nucleotide (nt) identity has been demonstrated in plants to be an important property responsible for directing association of endogenous small RNAs with different AGO effector proteins. However, it has not been investigated whether the 5' nt identity is an efficacious determinant for topically-applied chemically synthesized siRNAs. In this study, we employed a sandpaper abrasion method to study the silencing efficacies of topically-applied 21 base-pair siRNA duplexes. The *MAGNESIUM CHELATASE* and *GREEN FLUORESCENT PROTEIN* genes were selected as endogenous and transgenic gene targets, respectively, to assess the molecular and phenotypic effects of gene silencing. Collections of siRNA variants with different 5' nt identities and different pairing states between the 5' antisense nt and its match in the sense strand of the siRNA duplex were tested for their silencing efficacy. Our results suggest a flexibility in the 5' nt requirement for topically applied siRNA duplexes *in planta* and highlight the similarity of 5' thermodynamic rules governing topical siRNA efficacy across plants and animals.

## Introduction

RNA interference (RNAi) is one mechanism that organisms use to reduce the expression of specific target genes [1, 2]. Key features of the RNAi mechanism include the production and presence of small interfering RNA molecules (siRNAs) as part of an RNA-induced silencing

data are within the paper and its Supporting information files.

**Funding:** Monsanto Company (now Bayer Crop Science) provided support in the form of salaries for all authors as well as logistical support for the studies. The funder reviewed and approved the manuscript for publication but did not play a direct role in the study design, data collection and analysis, or preparation of the manuscript. The specific roles of authors are detailed in the 'author contribution' section.

**Competing interests:** All authors were employed by Monsanto Company (now Bayer Crop Science) while conducting this research. This study was funded by Monsanto Company. This does not alter our adherence to PLOS ONE policies on sharing data and materials.

complex (RISC) to suppress target RNA expression in a sequence specific manner [3, 4]. The siRNA duplexes are often produced by the endo-nucleolytic cleavage of longer double-stranded RNA (dsRNA) molecules by the proteins Dicer and Drosha (animals) or Dicer-Like (DCL; plants) [5]. siRNA duplexes are generally 20–25 nt in length, have 2nt 3' overhangs and are bounded by 5'-phospate and 3' hydroxyl moieties and additionally have 2'-O-methylation at their 3' ends [6, 7]. AGO proteins and the associated machinery are responsible for selectively sorting these siRNA duplexes and retaining one strand of the siRNA duplex. This retained strand is responsible for guiding the RISC complex to a target mRNA using sRNA sequence specificity and then mediates gene suppression [2, 8, 9].

In plant biotechnology, RNAi has been utilized to improve crop quality or provide protection by establishing stable transgenics expressing antisense strand or long dsRNA or by applying topically dsRNA as a biopesticide [10–12]. Topical application of long dsRNAs has been achieved by infiltration of agrobacterium, delivered as crude bacteria extracts, and using carrier peptides [10–12]. Following the biolistic delivery of siRNAs with gene silencing [13], multiple approaches to successfully topically deliver the shorter siRNAs were developed, *e.g.*, physical force, biological stress, lipid-based reagents, carbon dots and clay nanosheet technologies [2, 14–18].

Key steps in achieving siRNA efficacy are the association of an siRNA duplex with AGO effector protein followed by retention of one strand of the siRNA duplex to form a functional RISC complex [5, 9, 19, 20]. The retained siRNA strand is responsible for sequence-specific targeting of the mRNA and can be referred to as the guide strand or the antisense strand (AS) of the duplex. For consistency and clarity, we will refer to the targeting siRNA strand as the (AS) strand in this paper.

Multiple factors influencing AGO strand selection and sorting include the siRNA duplex structure and the 5' nucleotide identity of the AS strand [21–27]. In Drosophila, the duplex structure is a major determinant for miRNAs and siRNAs sorting into Ago1 and Ago2, respectively [24, 28]. In Arabidopsis, the 5' nucleotide of the AS strand plays a key role in sorting siRNA into specific AGOs [25, 27, 29]. AtAGO1, AtAGO2, AtAGO4 and AtAGO5 have verified 5'-nucleotide biases. Arabidopsis AGO1 preferentially binds miRNAs with a 5' uracil (U), AGO2 and AGO4 favor siRNAs with a 5' adenosine (A), AGO5 has a bias toward siRNAs with a 5' cytosine (C), whereas AGO4 primarily associates with heterochromatic-siRNAs bearing a 5' A [25, 27, 29, 30]. Interestingly, in the reports documenting AGO 5'nucleotide biases, no AGO preferentially bound siRNAs with a 5' AS guanosine (G). Introduction of an antisense 5'G perfectly paired with the sense strand of the duplex was shown to disable the siRNA's silencing efficacy [9].

siRNA strand selection and retention by an AGO can be mediated by the differential thermodynamic stability between the siRNA duplex ends: the duplex strand with the relatively less stably paired 5' end will be preferentially selected for AGO retention as matter of loading geometry [31, 32]. The thermodynamic properties of siRNA duplexes have been studied extensively and can be calculated as the differential free energy between two ends of the duplex and is represented by ΔΔG scores. Modification of ΔΔG scores in siRNA duplexes has been shown to affect siRNA efficacy. In animal systems, the introduction of a mismatch between the 5' terminal AS nucleotide and its normally paired nucleotide on the sense strand rendered an siRNA efficacious because it promoted retention of a targeting AS strand of the duplex within the AGO effector complex [32–35]. Mismatched terminal nucleotide modifications have not been tested directly in plant-based RNA silencing systems.

To understand the effect of the 5' AS nucleotide identity on gene silencing in plants, we employed a sandpaper abrasion method to topically deliver siRNA duplexes onto *N. benthamiana* leaves. The siRNA duplexes were designed to induce silencing of either an endogenous

**Table 1. Summary of all the siRNA sequences with their locations and the targeted genes, with the calculated ΔΔG.** siRNA are written as 5' ->3' sense strand: 5'->3' antisense strand. The bold and underlined letters denote non-native nucleotides. Pos, Position on the cDNA.

| SIRNA | SEQUENCE | GENE | POS | ΔΔG |
|---|---|---|---|---|
| CHL-1 WT | CGAAGGAGUUAUGCGAAUACC : UAUUCGCAUAACUCCUUCGUU | CHLH | 2191 | -1.80 |
| CHL-1 MT | CGAAGGA**CAA**AUGCGAAUACC : UAUUCGCAU**UUG**UCCUUCGUU | CHLH | 2191 | -1.80 |
| GFP-1 WT | GCAUCAAAGCCAACUUCAAGA : UUGAAGUUGGCUUUGAUGCCG | GFP | 542 | -2.50 |
| GFP-1 MT | GCAUCAA**UCG**CAACUUCAAGA : UUGAAGUUG**CGA**UUGAUGCCG | GFP | 542 | -2.50 |
| RANDOMIZED** | CCGAGUAUAAUAAGCAGAACC : UUCUGCUUAUUAUACUCGGUC | N/A | N/A | -2.30 |
| GFP-1 AU | GCAUCAAAGCCAACUUCA**U**GA : **A**UGAAGUUGGCUUUGAUGCCG | GFP | 542 | -2.60 |
| GFP-1 GC | GCAUCAAAGCCAACUUCA**C**GA : **G**UGAAGUUGGCUUUGAUGCCG | GFP | 542 | -0.40 |
| GFP-1 CG | GCAUCAAAGCCAACUUCA**G**GA : **C**UGAAGUUGGCUUUGAUGCCG | GFP | 542 | -1.20 |
| GFP-1 GG | GCAUCAAAGCCAACUUCA**G**GA : **G**UGAAGUUGGCUUUGAUGCCG | GFP | 542 | -1.50 |
| CHL-1 UU | CGAAGGAGUUAUGCGAAU**U**CC : UAUUCGCAUAACUCCUUCGUU | CHLH | 2191 | -3.10 |
| CHL-1 UG | CGAAGGAGUUAUGCGAAU**G**CC : UAUUCGCAUAACUCCUUCGUU | CHLH | 2191 | -2.10 |
| CHL-1 UC | CGAAGGAGUUAUGCGAAU**C**CC : UAUUCGCAUAACUCCUUCGUU | CHLH | 2191 | -3.10 |
| CHL-1 AA | CGAAGGAGUUAUGCGAAUACC : **A**AUUCGCAUAACUCCUUCGUU | CHLH | 2191 | -2.40 |
| CHL-1 AU | CGAAGGAGUUAUGCGAAU**U**CC : **A**AUUCGCAUAACUCCUUCGUU | CHLH | 2191 | -2.40 |
| CHL-1 AG | CGAAGGAGUUAUGCGAAU**G**CC : **A**AUUCGCAUAACUCCUUCGUU | CHLH | 2191 | -2.40 |
| CHL-1 AC | CGAAGGAGUUAUGCGAAU**C**CC : **A**AUUCGCAUAACUCCUUCGUU | CHLH | 2191 | -3.00 |
| CHL-1 CA | CGAAGGAGUUAUGCGAAUACC : **C**AUUCGCAUAACUCCUUCGUU | CHLH | 2191 | -2.40 |
| CHL-1 CU | CGAAGGAGUUAUGCGAAU**U**CC : **C**AUUCGCAUAACUCCUUCGUU | CHLH | 2191 | -3.00 |
| CHL-1 CG | CGAAGGAGUUAUGCGAAU**G**CC : **C**AUUCGCAUAACUCCUUCGUU | CHLH | 2191 | -1.20 |
| CHL-1 CC | CGAAGGAGUUAUGCGAAU**C**CC : **C**AUUCGCAUAACUCCUUCGUU | CHLH | 2191 | -3.00 |
| CHL-1 GA | CGAAGGAGUUAUGCGAAUACC : **G**AUUCGCAUAACUCCUUCGUU | CHLH | 2191 | -2.30 |
| CHL-1 GU | CGAAGGAGUUAUGCGAAU**U**CC : **G**AUUCGCAUAACUCCUUCGUU | CHLH | 2191 | -1.80 |
| CHL-1 GG | CGAAGGAGUUAUGCGAAU**G**CC : **G**AUUCGCAUAACUCCUUCGUU | CHLH | 2191 | -2.30 |
| CHL-1 GC | CGAAGGAGUUAUGCGAAU**C**CC : **G**AUUCGCAUAACUCCUUCGUU | CHLH | 2191 | -0.30 |
| CHL-2 wt | UCAUCUCCUCAUACCAAUCUU : GAUUGGUAUGAGGAGAUGAGC | CHLH | 2277 | -1.00 |
| CHL-2 GG | UCAUCUCCUCAUACCAAU**G**UU : GAUUGGUAUGAGGAGAUGAGC | CHLH | 2277 | -3.40 |
| CHL-2 UA | UCAUCUCCUCAUACCAAU**A**UU : **U**AUUGGUAUGAGGAGAUGAGC | CHLH | 2277 | -2.80 |
| CHL-3 WT | GAAGGAGUUAUGCGAAUACCA : GUAUUCGCAUAACUCCUUCGU | CHLH | 2190 | -3.70 |
| CHL-3 GG | GAAGGAGUUAUGCGAAUA**G**CA : GUAUUCGCAUAACUCCUUCGU | CHLH | 2190 | -5.60 |
| CHL-3 UA | GAAGGAGUUAUGCGAAUA**A**CA : **U**UAUUCGCAUAACUCCUUCGU | CHLH | 2190 | -5.20 |
| CHL-4 WT | GCCAACAGAUUGUGAACUCUA : GAGUUCACAAUCUGUUGGCCA | CHLH | 2316 | -1.60 |
| CHL-4 GG | GCCAACAGAUUGUGAACU**G**UA : GAGUUCACAAUCUGUUGGCCA | CHLH | 2316 | -4.00 |
| CHL-4 UA | GCCAACAGAUUGUGAACU**A**UA : **U**AGUUCACAAUCUGUUGGCCA | CHLH | 2316 | -3.40 |

** RANDOMIZED RNA is a non-targeting siRNA with no homology to any mRNA sequence. It is used as a negative control for all the experiments.

gene *MAGNESIUM CHELATASE SUBUNIT H* (*CHLH*) or the transgene *GREEN FLUORES-CENT PROTEIN* (*GFP*) [36]. This topical system allowed us to study the effect of the 5' AS nucleotide identity in RNA silencing. Our observations suggested that the thermodynamic asymmetry without specific regard to the 5' AS nucleotide identity plays an important role in the RNAi activity of topically applied siRNAs in plants.

## Materials and methods

All siRNA duplexes (Table 1) for topical applications and DNA oligos for Taqman assays (S1 Table) were synthesized by Integrated DNA Technologies (Coralville, Iowa, USA). Anti-

GFP antibodies (catalog # 632592) were sourced from Takara Bio USA Inc. *N. benthamiana* 16C- GFP line was described in Ruiz *et al.* [37]. The CHLH and the GFP sequences can be found in the *N. benthamiana* genome (https://solgenomics.net/organism/Nicotiana_benthamiana/genome) as Niben101Scf04388g00011.1 (or in the transcriptome collection (http://sefapps02.qut.edu.au/) as Nbv6.1trP75862) and in GenBank (https://www.ncbi.nlm.nih.gov/) as U87974.1, respectively.

### siRNA designer and ΔΔG calculations

The siRNAs to silence the target sequences (*GFP* or *CHL-H*) were selected with the following criteria (ΔΔG< = 0, Reynolds score > = 5, GC% as 40–60%, minimum free energy closes to 0) for 19mers [38–41], and extended with the native sequence as the 21 nt sRNAs with the 2nt 3' overhangs. Reynolds score [40] is an algorithm incorporating 8 characteristics associated with siRNA functionality, including low G/C content, a bias towards low internal stability at the sense strand 3'-terminus, lack of inverted repeat, and sense strand base preferences (positions 3, 10, 13 and 19).

The terminal stability ($\Delta G$, kcal. mole$^{-1}$) of the siRNA duplex was calculated using mfold by summing the nearest-neighbor contributions for the first five nucleotides (four nearest-neighbor energies) at the 5' end plus the free ssRNA overhangs, using the weblink (http://www.unafold.org/Dinamelt/applications/two-state-melting-folding.php) for the calculation [42]. Differential end stability ($\Delta\Delta G$, kcal·mol−1) was calculated as the difference in thermodynamic stabilities at each end.

### Growth conditions and sandpaper RNA application treatment

Both *N. benthamiana* wild type and 16C-GFP transgenic plants were established in 4-inch pots and grown for two weeks in a controlled environment growth chamber (263 μmol m-2 s-1 of light intensity, set at 14 hours of light at 23 ± 1˚C and 10 h dark at 18 ± 1˚C). The lyophilized siRNA powder from IDT was resuspended in $H_2O$ at a desired concentration. 10–20 μl of dsRNA was applied to the third and/or fourth *N. benthamiana* leaf. After application, the solution was gently spread on the top of the leaf using a pipette tip. P600 sandpaper was taped around a small cylinder, and was rolled over the top of the leaf, causing abrasion. The application leaves were allowed to dry, and plants were grown for an additional four days in the controlled environment growth chamber prior to being photographed under white light (for *CHLH* siRNA treatment) or with UV light (for *GFP* siRNA treatments). Six plants were included for each treatment. Tissue from the application regions was harvested for Taqman assay or western blots using a 4mm disposable biopsy punch (Sklar Instruments, West Chester, PA), was frozen immediately on dry ice and was stored for analysis.

### GFP protein quantification by western blot

To determine GFP expression in leaves treated with siRNA, two 4 mm leaf discs were placed in FastPrep Lysing Matrix D tubes (Fisher Scientific) with 150 μl extraction buffer (20 mM Tris pH 7. 5, 5 mM $MgCl_2$, 300 mM NaCl, 5 mM DTT, 0. 1% NP-40, 1X cOmplete protease inhibitor cocktail (catalog # 04693159001, Roche) and subjected to shaking for 20 seconds at 4 m/s. Fast prep tubes were then centrifuged at 4˚C for 10 min and equal amounts of extract was diluted with Novex NuPAGE LDS 2X Sample Buffer (supplemented with 10% B-mercaptoethanol) to a 1X final concentration. Proteins were denatured at 95˚C for 10 min before loading onto a 7.5% BioRad TGX gel. After the gel was run, the proteins were transferred to nitrocellulose using a turbo blotter (Bio-Rad). The membrane was blocked with 5% blotto in TBST (Thermo Fisher Scientific), and anti-GFP antibody (catalog # 632593, Takara Bio USA

Inc) was incubated for 1 hr at 1:2000 in 5% blotto. The membrane was then washed 3 times (5 min each) in TBST and then incubated for 1hr in 1:15000 goat anti-rabbit HRP-conjugated secondary antibody. The membrane was then washed 4 times (10 min each) in TBST and exposed in Pierce Super Signal Femto (Thermo Fisher Scientific) for 1 min. An automatic exposure was done on a BioRad imager. Ponceau staining was completed for 2 min in 1X Ponceau-S and imaged with the BioRad imager on the colorimetric setting.

### *GFP* and *CHLH* transcript quantification

For each sample, three 4 mm frozen leaf discs were ground with ball bearings and extracted using Direct-zol™-96 RNA (Zymo Research; Irvine, CA). Total RNA was eluted in DNAse/RNAse free water, and used directly for cDNA synthesis. In the high throughput laboratory, the set amount of leaf samples routinely have RNA yields within the similar range, and therefore, only a few wells were spot-checked to ensure the concentration was less than 200 ng/ul to be below the saturating level of reverse transcription kit, Oligo dT cDNA was generated with High Capacity cDNA Reverse Transcription Kit (Applied Biosystems; Foster City, CA) per the manufacturer's protocol. Singleplex qPCR was performed with a 4-fold dilution of cDNA using either Quanta Bio (Beverly, MA) PerfeCTa Fast Mix II for probe based assays (EF1a and UKN1) or PerfeCTa SYBR green Supermix (CHLH and GFP) and run on BioRad CFX96 Real-Time PCR Detection System (Hercules,CA). For quantification, EF1a and UKN1 was used as internal normalizer genes. The primers and probes designed for *EF1a*, *UKN1*, *GFP* and *CHLH* can be found in S1 Table. Relative Quantity (RQ) was calculated by 2^-(GOI Ct–GeoMean Internal Normalizer Ct). Standard Relative Quantity was then calculated to rescale values onto a scale of 100 by dividing the RQ by the mean of the baseline treatment (sandpaper only without any siRNA). Statistical analysis was performed and resulting LS means and error bars were plotted as a bar chart.

### Image capture

Plant leaf images were captured using the Canon EOS70D camera. White and fluorescent light was projected onto the leaves using LED lights (Model: SL3500-D) from Photon System Instruments (Drasov, Czech Republic). For GFP fluorescence capture, the lights were adjusted to emit a wavelength of 447 nm (appeared as blue light) and a 460 nm filter was placed on the camera.

Images were analyzed using the ImageJ software [43]. Using ImageJ, total leaf area was calculated. Utilizing the color differences between non-silenced and silenced tissue (green vs yellow for *CHLH* silencing; green vs red for *GFP* silencing), the total area of silencing was determined. To maintain consistency, the threshold that determined silenced tissue vs non-silenced tissue remained constant for all images taken for an experimental comparison. The percent silencing was then calculated by dividing the total silencing area by the total leaf area.

### Statistical analysis

The statistical assessment of differences were conducted for bar charts of all data collections. One-way ANOVA and Fisher's LSD test was used for RNA quantitation, protein quantitation by western blots, and sandpaper images analysis with the single factor as 5' nucleotide pair. The data of the siRNA comparison in the protoplast reporter systems was analyzed using a two-way ANOVA and Fisher's LSD test with 5'pair and sRNA rate as the factors. In all cases, means not followed by the same letter indicate statistical significance ($\alpha = 0.05$).

**Protoplast GFP reporter assay of siRNA activity**

Plasmids were prepared using the Qiagene maxi-prep kit (Qiagen, Hilden, Germany). The cDNA sequence encoding full length GFP was cloned downstream of a CaMV 35S promoter. The short-fragment of "5'-CGAAGGAGTTATGCGAATACC-3'" matching part of the *CHLH* transcript was inserted at the 3' end of *GFP* sequence to be used as a reporter system for GFP expression when co-transfected with siRNAs to the protoplast.

Protoplasts were prepared from whole leaves of 8-11-day-old *N. benthamiana* plants after 3 days etiolation in the dark using an enzyme solution containing cellulase R-10 and macerozyme R-10 (Yakult Pharmaceuticals, Japan) to remove cell walls [44]. PEG-assisted transformation reactions contained 2 μg of a GFP-expressing plasmid (containing target sequences) and either 3, 0.75, 0.19, or 0.06 μg of dsRNA, plus transformation controls containing 0.65 μg of Firefly luciferase and 0.33 μg of Renilla luciferase plasmids. Control wells had either GFP plasmid only, or no DNA input. Master aliquots of 10 μL were placed into a separate Falcon 353910 plate(s), in quadruplicate. Post-transformation protoplasts were resuspended with 180 μL Buffer containing 0.025% BSA and 5 μg/mL streptomycin and incubated overnight at room temperature. The following day, a 25 μL aliquot of the incubation mixture was used for luciferase assays using a modified protocol of the Dual-Glo system (Catalog # E2980, Promega Inc) to monitor the protoplast healthiness and transfection effectiveness. Another 25 μL aliquot of the incubation mixture was used for GFP expression analysis on an Operetta high content imager (Perkin-Elmar). The GFP Intensity Sum per well was calculated for each sample and averaged.

## Results

### Development of a topical application method for siRNA delivery to *N. benthamiana* leaves

To directly assess the gene silencing effects of siRNAs at both phenotypic and molecular levels, we employed a sandpaper abrasion method to deliver siRNAs to leaves of young *N. benthamiana* plants. The endogenous gene, *CHLH*, was selected as a candidate because of the previously reported chlorotic phenotypes of its silencing mutant. To incorporate heme required for the synthesis of chlorophyll, the magnesium-protoporphyrin IX chelatase complex consists of three subunits (D, I, and H), and distributes protoporphyrin IX into the branched tetrapyrrolic pathway. Transgenic tobacco (*Nicotiana tabacum*) plants expressing antisense *CHLH* RNA became deficient in chlorophyll biosynthesis and showed a chlorotic (yellow-spotted) phenotype [45]. A wildtype siRNA duplex (CHL-1 WT, Table 1) with a core 19 mer and an antisense sequence of 5' UAUUCGCAUAACUCCUUCG plus 3' 2nt overhangs was selected to target the *CHLH* mRNA (Fig 1A). As a negative control, a mutant version (CHL-1 MT, Table 1) was designed by introducing a 3nt substitution at position 10–12 of the antisense strand (AAC -> UUG), to abolish the cleavage activity [46]. Both the wildtype and mutant CHLH siRNAs were formulated in H$_2$O and applied to the wildtype *N. benthamiana* using sandpaper. At 4 days post application (dpa), control leaves applied with mutant or a non-targeting randomized siRNA lacked chlorotic spots. In contrast, leaves treated with 21nt dsRNA duplexes targeting *CHLH* demonstrated chlorotic spots (covering about 10% of the total leaf area), presumably resulting from the interference with chlorophyll assembly after down-regulation of *CHLH*.

We additionally tested whether the sandpaper delivery method could be used to effectively silence the GFP transgene present in the 16C line of *N. benthamiana* [47]. GFP emits green fluorescence light that is observed under ultraviolet (UV) light activation. However, when the GFP protein is absent, red chlorophyll autofluorescence is observed under UV light. A 21bp

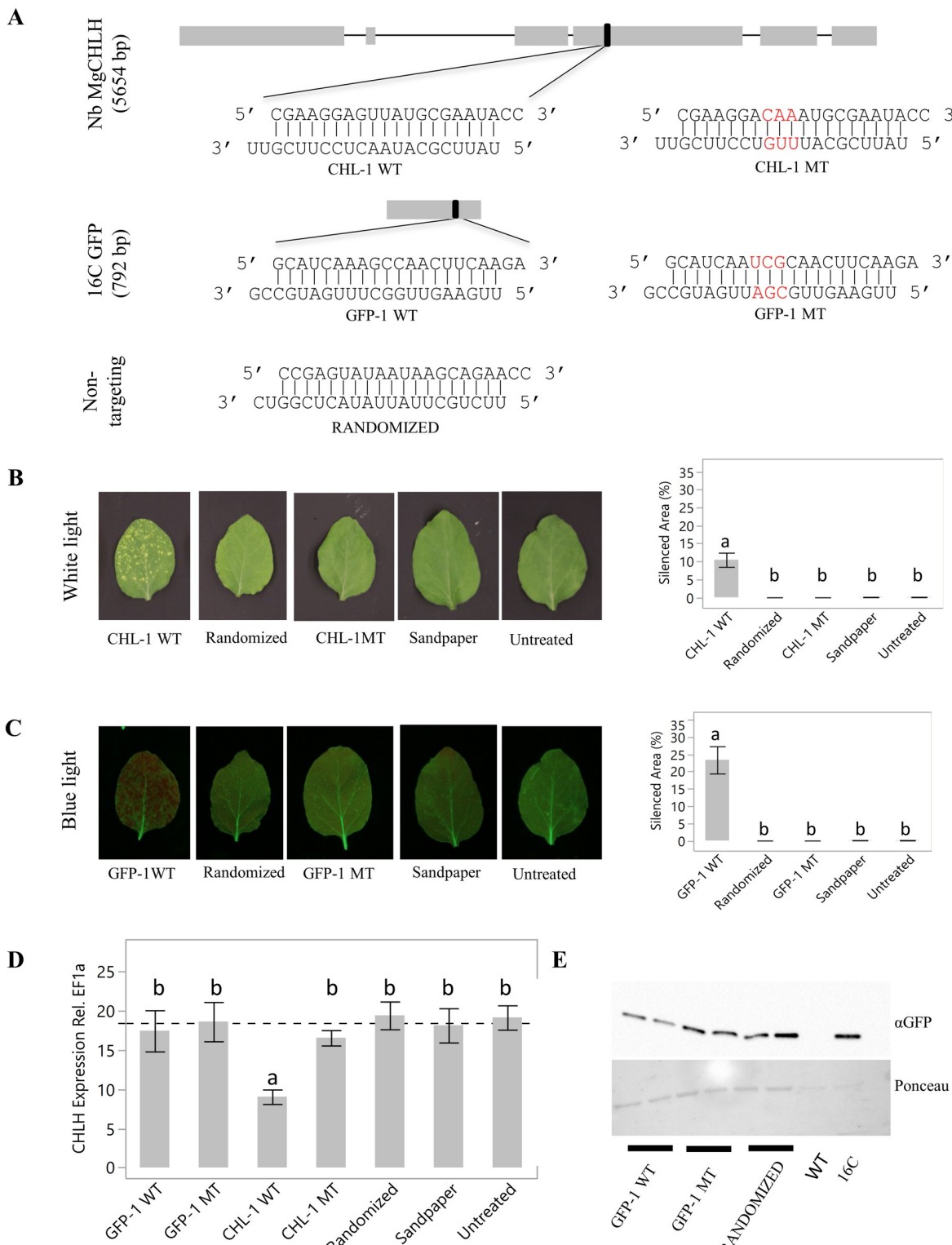

**Fig 1. Sandpaper application of siRNA duplexes resulted in an RNAi phenotype.** A, Location of siRNAs targeting the *CHLH* and *GFP* genomic locations in the *N. benthamiana* wildtype and16C lines. Boxes and lines indicate the exons and introns, respectively. The 21 bp sequences with 2 nt 3' overhangs were tested as wild type or with mutations shown in red. The top strand is the sense sequence, and the bottom is the antisense sequence of the duplex. Leaves were treated with the indicated siRNA duplexes by sandpaper abrasion and photographed under white (B) or blue light (C) 4 dpa. The phenotypic area was quantified with Image J and was graphed as percent of the

treated leaf area; error bars represent standard error of the means. D, *CHLH* transcript levels were determined in leaves harvested 4 dpa; error bars represent standard error of the means. E, GFP protein levels were determined by western blot from leaf tissues collected 4 dpa. Means not followed by the same letter indicate statistical significance ($\alpha$ = 0.05).

siRNA sequence (GFP-1 WT, Table 1) with a 19bp core and antisense strand of 5' UUGAA GUUGGCUUUGAUGC was selected to target the *GFP* transgene (Fig 1A). Similarly, mutations were introduced at antisense positions 10–12 to disable siRNA efficacy (GFP-1 MT, Table 1). Leaves subjected to topical sandpaper delivery of dsRNA GFP siRNAs displayed red chlorophyll autofluorescence covering approximately 25% of the total leaf area and demonstrating a lack of GFP protein activity. In contrast, leaves treated with either mutant or non-targeting sequences showed uniform GFP fluorescence under UV light (Fig 1C and S6 Fig). As was observed for *CHLH*, disruption of the 10–12 nt of the antisense strand inhibited the red fluorescence phenotype, suggesting these observations resulted from RNAi activity (Fig 1C and S6 Fig).

To determine if the observed phenotypes correlated with reduced mRNA levels for the targeted genes, quantitative RT-PCR assays were performed on RNA isolated from leaf punches collected in the phenotypic areas. As expected, treatment with the wildtype siRNA targeting *CHLH* (CHL-1 WT) resulted in an approximate 50% decrease in *CHLH* mRNA in comparison to non-treated controls. No reduction in mRNA levels were observed after treatment with either the mutant (CHL-1 MT) or the non-targeting sequence (RANDOMIZED) (Fig 1D). No difference in *GFP* mRNA levels were observed after treatment with GFP siRNAs in comparison to untreated GFP controls (S2 Fig). To assess whether the observed GFP phenotypes are from reduced GFP protein expression, protein extracts were subjected to western blot analysis using an anti-GFP antibody. As shown in Fig 1E, the GFP protein level was reduced in samples from leaves treated with the wildtype GFP siRNA, but not with the GFP mutant (GFP-1 MT) or the non-targeting siRNA (RANDOMIZED).

To assess the relationship between siRNA dose and the intensity of phenotype, leaves were treated with siRNA at two concentrations, 0.5 and 2 µg/µl. As shown in S1 Fig, for both *CHLH* (CHL-1 WT) and *GFP* (GFP-1 WT), silencing phenotypes were enhanced with an increased dose of the siRNA. No phenotypes were observed with the randomized control siRNA sequences in either doses.

These results support the conclusion that the phenotypes generated using the topical sandpaper RNA delivery method are the result of RNAi. We therefore used this topical sandpaper abrasion method to assess the effect of siRNA designs on gene silencing activity.

## Effects of siRNAs with variants at the 5' terminus of antisense strand on gene silencing

To understand the 5'nt requirement for siRNA activity after topical application, four variants of the *GFP* siRNA were designed by modifying the 5' nt of the antisense strand, and its corresponding pair on the sense strand (Fig 2 and S7 Fig). The variants tested were U-A, A-U, G-C, and C-G pairs (Table 1), where the first letter denotes the 5' AS nucleotide identity (position 1 of the AS strand) and the second letter denotes its 'pair' in the opposite strand of the duplex (position 19 of the sense strand). Leaves treated with the native U-A pairing displayed the red spotted phenotype, resulting from loss of GFP expression. A similar phenotype was observed for both C-G and A-U variants, while the leaves treated with G-C variants did not exhibit a red-spotted phenotype. To test whether the introduction of 5'G as the first nt of the antisense strand was the main factor leading to the disappearance of the phenotypes, an additional

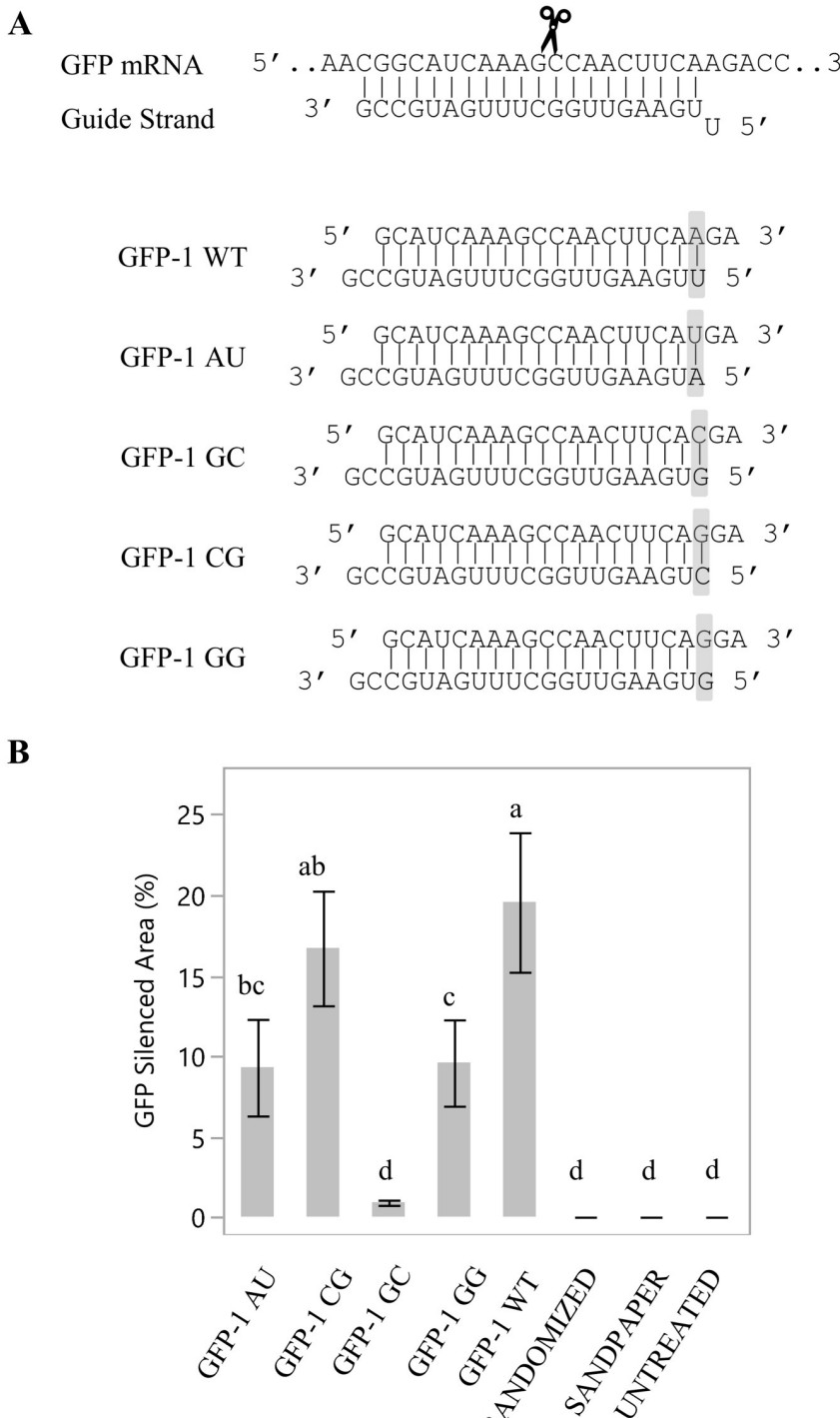

**Fig 2. The influence of the 5' nucleotide of the antisense strand of a 21nt siRNA duplex on *GFP* transgene silencing.** A, Illustration of interactions between the *GFP* mRNA and the AS strand of siRNA duplex. dsRNAs with different 5' nucleotides for the antisense strand (paired with a complementary or identical (G) nucleotide at the sense strand) were tested. B, *GFP* silenced area was quantified and expressed as percent of the treated leaf area; error bars are standard error of the means. Means not followed by the same letter indicate statistical significance (α = 0.05).

variant was made which contained the 5'G as the first nt on the antisense strand mismatched with a non-pairing G on the 19th position of the sense strand. This G-G mismatch variant displayed the red spotted *GFP* silencing phenotype, at statistically the same level to the leaves treated with the wildtype siRNA. This observation suggests that siRNAs with a 5'G on the antisense strand should not be excluded from siRNA testing when used for topical application.

To further investigate the antisense strand's first nucleotide pairing state on RNAi efficacy, additional siRNAs targeting *CHLH* were designed and their efficacy tested (Fig 3, S3 and S8 Figs). From the wildtype siRNA (CHL-1 WT) targeting *CHLH*, an additional 15 CHL-1 dsRNA variants that included all 5' antisense strand pairing states of G, A, U, and C (Table 1) were designed. The ΔΔG for each of the siRNAs was calculated based on the difference in free energy for each end of the duplex calculated from the stacking energy of the first 5 pairs of the sequence and the free end loops. The calculated ΔΔG for all variants were below 0, including the duplex CHL-1 GC with a 5'G on the antisense strand paired with a C on the sense strand (Table 1).

Among the plants treated with CHL-1 variants with perfect pairing at the 5' AS end, CHL-1 WT siRNA (5' U-A), CHL-1 AU, and CHL-1 CG all displayed the chlorotic phenotypes. No chlorotic phenotype was observed for the CHL-1 GC variant (Fig 3 and S8 Fig). Compared to the *CHLH* mRNA level in the control treatments (Sandpaper only, untreated, and GFP siRNA), varied degrees of transcript reduction of *CHLH* were detected in all the variants except the CHL-1 GC (S3 Fig). Treatments with CHL-1 variants with mismatches between the 5' AS nucleotide and its sense pair presented chlorotic spots of varied intensity, suggesting they are capable of knocking down the *CHLH* transcript to produce a phenotype. Introduction of a mismatched state at the 5' antisense strand in the siRNA allows for activity of 5'G designed siRNA.

## RNAi effects by siRNAs variants in protoplasts

To quantify dose effects of the siRNA variants in RNA silencing, we used a protoplast transient assay [48, 49]. A siRNA reporter plasmid containing a *GFP* gene that had a short stretch of *CHLH* sequence (5'-CGAAGGAGTTATGCGAATACC-3') embedded in the 3' end immediately after the translational stop codon was constructed to test siRNA efficacy (Fig 4 and S4 Fig). The plasmid and one of the sixteen variant siRNAs were co-transfected into *N. benthamiana* leaf protoplasts. An efficacious siRNA should recognize the embedded *CHLH* target sequence and elicit the RNAi gene silencing pathway, resulting in a decrease of GFP fluorescence signal. The optimal amount of plasmid for the assay was determined to be 2 μg by a titration experiment (S5 Fig). All 16 variants of siRNAs targeting *CHLH* were applied at 0.06, 0.19, 0.75 and 3 μg per sample. With the increased amount of siRNA, a reduction of GFP signal was observed for all the variant siRNAs applied (Fig 4). As was seen for the topically applied sample, the siRNA containing a 5' G-C on the antisense strand did not suppress the GFP fluorescence as efficiently as all the other variants. At 0.19 μg siRNA, the intensity of GFP fluorescence was similar as those after treatment with the randomized negative control (Fig 4B). Even at the highest amount tested (3 μg), the GFP fluorescence level remained at about 80% of the negative control. This observation, consistent with data in Fig 3, supports that silencing by an siRNA starting with 5'G-C on the antisense strand is inefficient. In contrast, the mismatched siRNA variants beginning with 5' AS G all showed significant efficacy compared to controls.

## Activation of siRNA by introducing 5' mismatch nucleotides

The native sequences tested so far were selected having a 5'U on the antisense strand as one of the initial design criteria. We therefore asked whether a siRNA sequence beginning with a

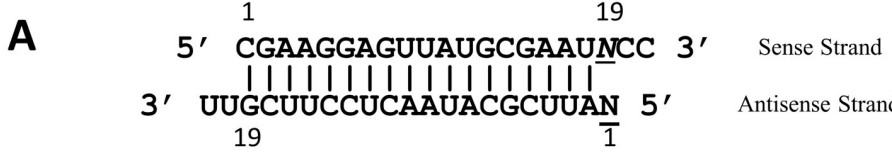

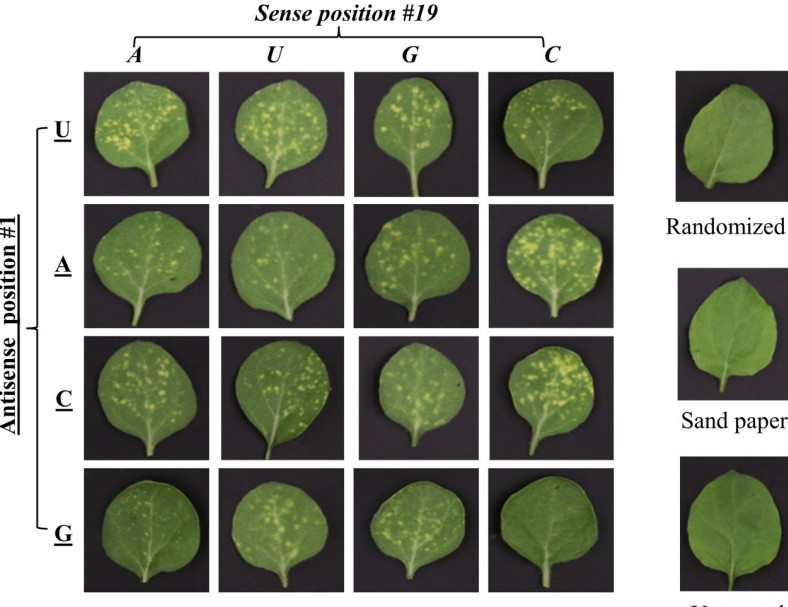

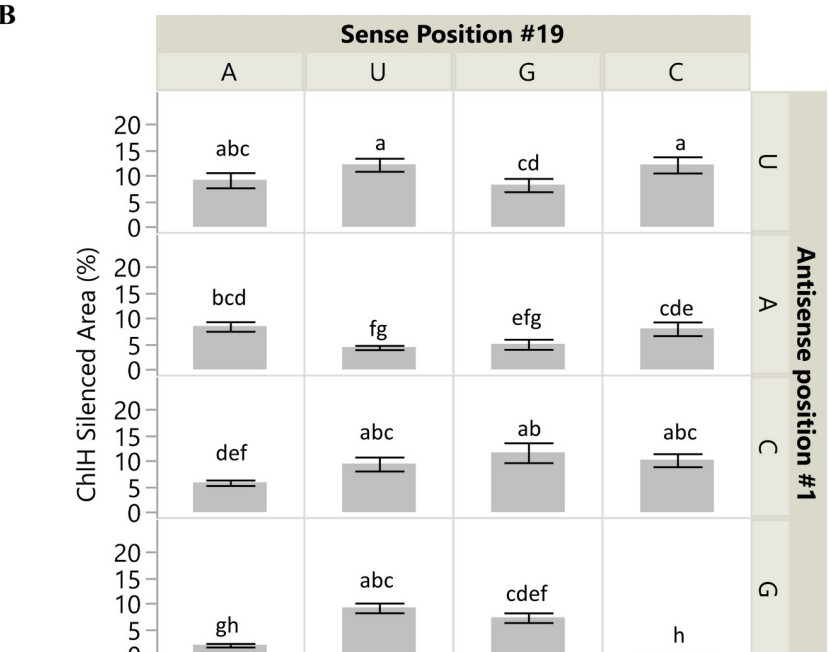

**Fig 3. The influence of the 5' nucleotide of a 21 bp dsRNA on silencing the *CHLH* gene.** A, Each 5' AS nucleotide was tested both in a paired state and in an unpaired states with all other nucleotide possibilities. Leaves were harvested 4 dpa and photographed under white light. B, Phenotypic area was quantified and expressed as percent of the treated leaf area. Means not followed by the same letter indicate statistical significance ($\alpha = 0.05$).

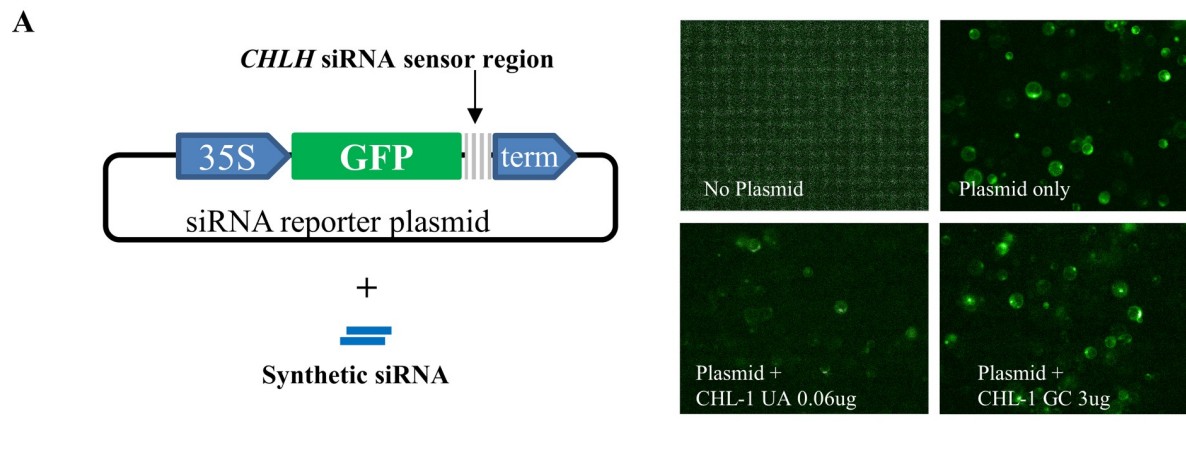

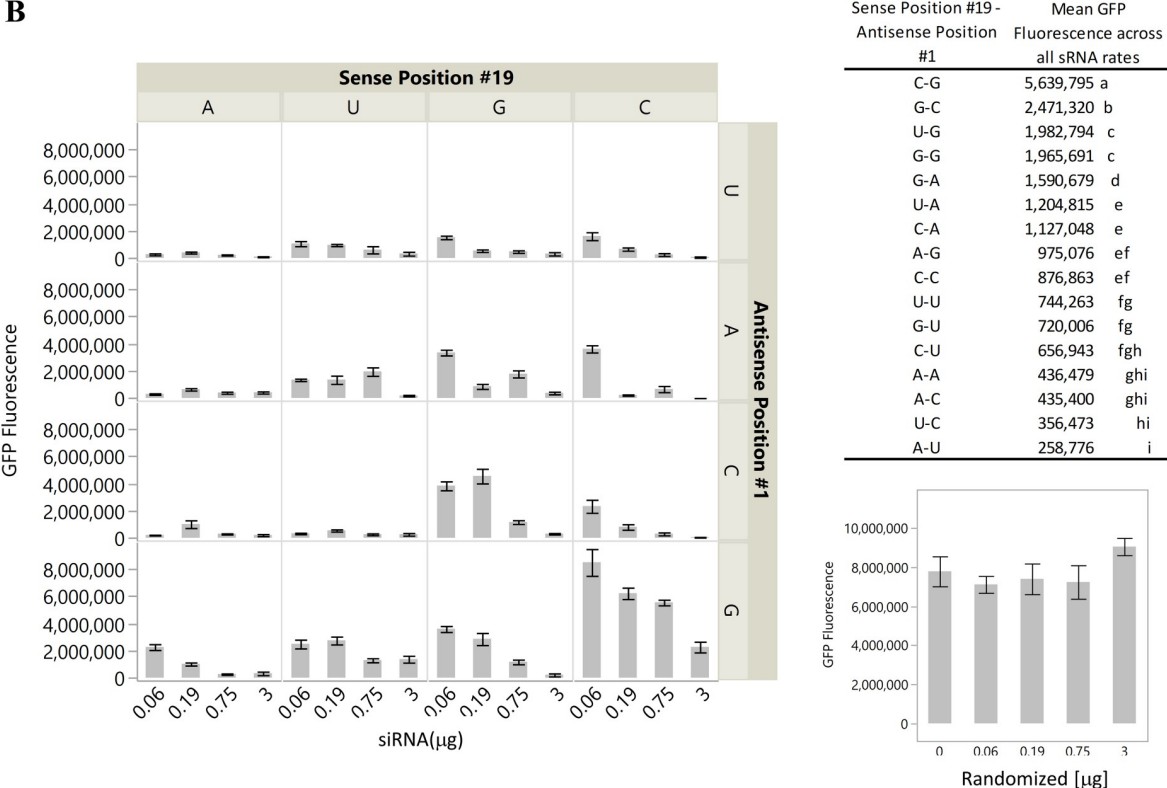

**Fig 4. Silencing of a *GFP*:*CHLH*-fragment fusion in protoplasts was used to measure the efficiency of silencing for sixteen *CHLH* siRNA variants.** A, Diagram of the siRNA reporter plasmid containing the *GFP*:*CHLH* fragment:3' UTR cassette for gene suppression in protoplasts, the synthetic siRNAs, and the representative protoplast images captured by Oppereta imager of no plasmid, plasmid only, plasmid + CHL-1 UA, and plasmid + CHL-1 GC. B, siRNAs as described in the legend to Fig 3 were transfected together with the *GFP*:*CHLH*-fragment fusion, and GFP fluorescence was quantified at 525 nm with 488 nm excitation. Means not followed by the same letter indicate statistical significance ($\alpha = 0.05$).

native 5'G was capable of reducing gene expression and whether an introduction of 5' terminal mismatches or 5' U substitutions could activate or enhance any silencing properties. Three different siRNAs were selected in the middle region of the *CHLH* gene with the following criteria: GC % at between 40% to 50%, ΔΔG less than 0, Reynolds score >5, and oligo MFE < or equal

to 0 (Fig 5A and 5B). Two variant types were created for all three native siRNA duplexes. In the first set, a mismatched state was incorporated by substituting position 19 of the sense with a G, resulting in a G-G mismatch. In the second set of siRNA sequences, the 5' G on the AS strand was substituted with a 5' U and a compensatory A on the sense strand was also incorporated. As shown in Fig 5C and S9 Fig, no chlorotic phenotype was observed after treatment with the native sequences starting with a 5'G-C. In contrast, both the 5' G-G mismatch and 5' U-A substitution in all three variant siRNA groups led to chlorotic phenotypes.

## Discussion

In this study, we demonstrated that siRNA duplexes delivered into *N. benthamiana* leaves using the sandpaper abrasion method is an effective procedure for activity assessment of exogenously applied siRNA duplexes. Overall, both topically applied siRNAs and siRNAs transformed into plant protoplasts showed consistent gene silencing effects in terms of generation of a phenotype, mRNA or protein reduction, and loss of activity after incorporating slicer mutations into the siRNA sequences.

One known consideration relating to siRNA activity is sorting and incorporation into an AGO effector protein. Plants have an expanded family of diverse AGO proteins and the number of family members varies with the plant species [9, 50]. As examples, there are 9 AGO genes in *N. benthamiana* [51], 10 AGO genes in Arabidopsis [52] and 19 in rice [53]. These different AGOs have specialized functions in plant development, defense, and genome maintenance based on expression pattern and by selectively binding to different classes of siRNAs [54–57]. The AGO protein or proteins utilized with topical RNAi remains unknown. It is possible that multiple AGO proteins may be involved in gene silencing after topical application of siRNAs [2, 8, 25, 58, 59]. Further biochemical or genetic might help elucidate the AGO preference component of topically applied siRNAs.

The preference of a 5'U at the antisense strand of an efficacious siRNA has been reported in both plants and animals and widely used as a criterion for designing siRNA [25, 26, 39, 40, 60]. In plants, most of the endogenous miRNA populations have a 5' U and are produced by DCL from a precursor structure with base pairing at that 5' AS position. Among the 162 identified naturally occurring miRNA in *N. tabacum* (http://www.mirbase.org/), none of the mature miRNAs contains a 5' AS mismatch. Endogenous siRNAs produced through an RNA-dependent RNA polymerase would also have perfect base-pairing prior to DCL cleavage. The sandpaper based application method allowed for an opportunity to evaluate the specific base-pairing effects of 5' nt when exogenous siRNA molecules were topically applied.

The silencing efficacy between siRNA with 5' AS GC (non active) and the 5' CG (active) for both CHL-1 and GFP-1 variants may appear surprising when considering both versions have the similar pairing energy. We speculate this difference could result from the ΔΔGs that takes into consideration both the stacking energy and the pairing energy when using the nearest neighboring models. In fact, the CG versions for both CHL-1 and GFP-1 are both at -1.2 while that of the GC version are -0.3 and -0.4 for GHL-1 and GFP-1, respectively. An additional influencing factor could be the binding preference and affinity of the 1st nucleotide of the AS strand to the corresponding AGO protein [26].

The observation that most of the variant siRNAs designs we tested in relation to 5' nt identity and pairing states can be efficacious, either when applied topically or co-transfected with a reporter plasmid in protoplasts was intriguing. Furthermore, the ability to activate non-efficacious duplexes (GC at the 5' AS) through incorporation of mismatch states suggested that this property is an important one to consider during the siRNA design process. The modifications made to enhance or activate siRNA duplex activity would lower the 5' stability of the antisense

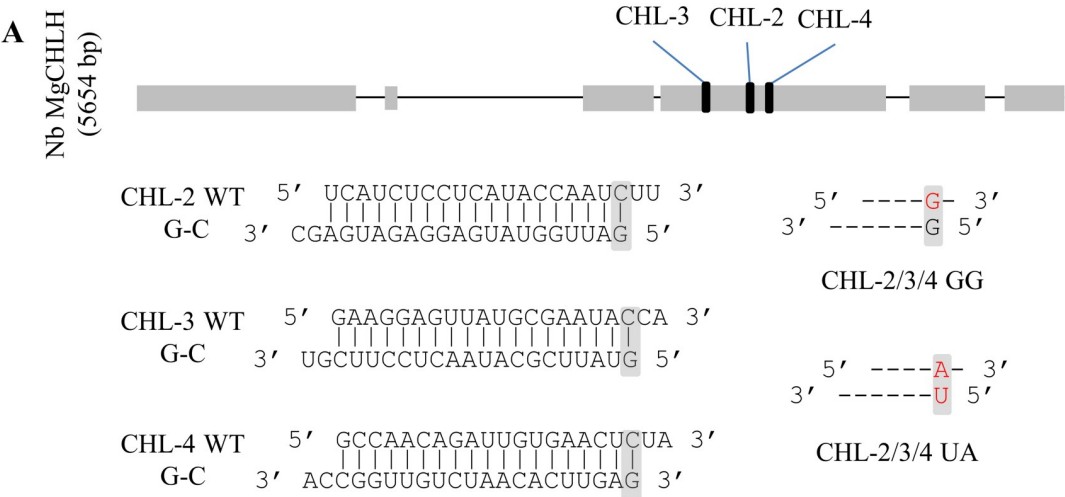

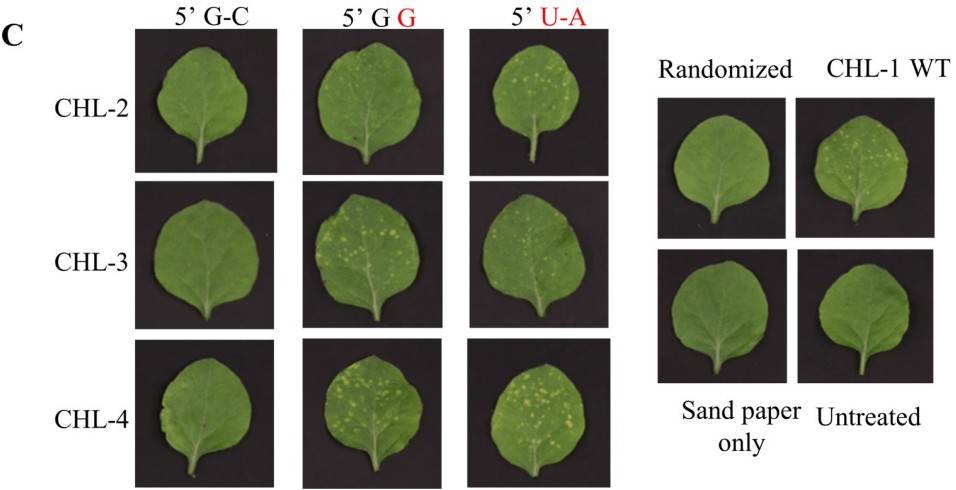

**Fig 5. siRNAs containing a 5' G can be efficacious for gene silencing when the nucleotide on the antisense strand is the non-complementary G or when a U-A pair is substituted.** A, Location of three 21nt siRNAs in the *CHLH* genomic sequence in the *N. benthamiana*. Boxes and lines indicate the exons and introns, respectively. These siRNAs were synthesized in their native form as a G-C pairing at the 5' end of AS strand or substituted with a G-G mismatch or U-A pairing. The nucleotides of 1st position of AS and 19th position of S strands are shaded in grey. B, Key properties for the siRNAs tested were compared (40, 41). C, siRNA duplexes were applied to *N. benthamiana* leaves with sandpaper abrasion, and treated leaves were photographed under white light 4 dpa.

end, potentially promoting the preferential incorporation of the correct targeting strand into the RISC complex [31, 32]. Previous studies have shown that strand selection is a fundamental step for the correct strand to load into the RISC complex [31, 40, 61]. These results highlight the critical step of 5' instability for strand selection and our finding suggests that the decreasing the relative stability of the 5' antisense end may override the nucleotide identity requirement for topically applied synthetic siRNAs. Reactivation of non-functional siRNAs by incorporation of a mismatch state within the duplex or with a non-native 5' U is also consistent with the role of the first nucleotide of the antisense strand not participating in seed region or slicing activity [26, 62].

In summary, this work demonstrates that all 5'AS nucleotides can be incorporated into an effective siRNA design for topical or *in planta* application, providing a broader swath of efficacious siRNA sequences for gene silencing.

## Supporting information

**S1 Fig. Gene silencing by siRNAs applied using sandpaper abrasion was dose dependent.** A, Leaves were treated with the indicated triggers at 2 concentrations by sandpaper abrasion and were photographed 4 dpa under white or blue light. B, Phenotypic area was quantified with Image J and is graphed as percent of the treated leaf area; error bars represent standard error of the means.
(TIF)

**S2 Fig. GFP transcript levels of the leave discs harvested 4dpa after the sandpaper application of siRNA duplexes.** The error bars represent standard error of the means.
(TIF)

**S3 Fig. *CHLH* transcript level in leaves treated with siRNA duplexes.** *CHLH* mRNA levels were determined in leaves harvested 4 dpa; error bars represent standard error of the means. The asterisk denotes the treatment of the wildtype siRNA that starts with a 5'U on the antisense strand with a paired A on the sense strand.
(TIF)

**S4 Fig. Plasmid map of the siRNA GFP reporter plasmid with an embedded CHLH fragment at the 3' end.** The *GFP* sequence is followed by the short stretch of *CHLH* sequence (5'-CGAAGGAGTTATGCGAATACC-3'). The expression of GFP is driven by the CaMV promoter and the RbcS terminator.
(TIF)

**S5 Fig. The optimal plasmid concentration for GFP silencing in *N. benthamiana* protoplasts was determined by a titration experiment.** The reporter plasmid (pMON406238) was co-transfected at different concentrations into *N. benthamiana* protoplasts with 3 μg siRNA, and GFP fluorescence was determined after overnight incubation. RANDOMIZED, control siRNA.
(TIF)

**S6 Fig. Photographs of leaves treated with sandpaper application of siRNA duplexes.** Leaves were treated with the indicated triggers by sandpaper abrasion and photographed 4 dpa to observe silencing phenotypes. White rectangle boxes highlight the leaf in each treatment shown in Fig 1(A) and S1(B) Fig.
(TIF)

**S7 Fig. Photographs of leaves treated with sandpaper application of siRNA duplexes to silence GFP expression in Fig 2.** Leaves were treated with the indicated triggers by sandpaper abrasion and photographed under blue light 4 dpa to observe phenotypes.
(TIF)

**S8 Fig. Photographs of leaves treated with sandpaper application of siRNA duplexes to silence CHLH expression in Fig 3.** Leaves were treated with the indicated triggers by sandpaper abrasion and photographed under white light 4 dpa to observe silencing phenotypes. White rectangle boxes highlight the leaf in each treatment shown in Fig 3.
(TIF)

**S9 Fig. Photographs of leaves treated with sandpaper application of siRNA duplexes to silence CHLH expression in Fig 5.** Leaves were treated with the indicated triggers by sandpaper abrasion and photographed under white light 4 days after treatment to observe silencing phenotypes. White rectangle boxes highlight the leaf in each treatment shown in Fig 5.
(TIF)

**S1 Table. RT-qPCR primer and probe sequences for transcript measurements (sequences are written as 5'-3').**
(DOCX)

**S1 Raw images.**
(PDF)

## Acknowledgments

The authors thank the controlled environment team of Emily Schuchardt, Sherry LaVallee, Dr. Chris Kavanaugh for plant growth and care, Baohua Wang for her help in identifying siRNAs starting with any nucleotides, and Dr. Brian Eads for his support on ΔΔG analysis of all the siRNA sequences. We also thank Drs. Ed Allen and Xiaoyu Liu for their critical reading and thoughtful inputs on the manuscript. Sequences of siRNAs GFP-1 WT were designed and tested originally by Dr. Sunny Gilbert.

## Author Contributions

**Conceptualization:** Peizhen Yang, Ericka Havecker, John Bradley.

**Data curation:** Peizhen Yang, Ericka Havecker, Matthew Bauer, Carl Diehl, Bill Hendrix, Paul Hoffer, Timothy Boyle, Amy Caruano-Yzermans.

**Formal analysis:** Peizhen Yang, Ericka Havecker, Bill Hendrix, Paul Hoffer, Amy Caruano-Yzermans.

**Investigation:** Peizhen Yang, Ericka Havecker, Matthew Bauer, Carl Diehl, Timothy Boyle, John Bradley, Amy Caruano-Yzermans.

**Methodology:** Peizhen Yang, Ericka Havecker, Matthew Bauer, Carl Diehl, Bill Hendrix, Timothy Boyle, John Bradley, Amy Caruano-Yzermans.

**Project administration:** Peizhen Yang.

**Resources:** Jill Deikman.

**Software:** Paul Hoffer.

**Supervision:** Peizhen Yang, Bill Hendrix, Jill Deikman.

**Validation:** Peizhen Yang, Ericka Havecker, Matthew Bauer, Bill Hendrix, Timothy Boyle.

**Visualization:** Ericka Havecker, Matthew Bauer, Bill Hendrix, Paul Hoffer.

**Writing – original draft:** Peizhen Yang, Jill Deikman.

**Writing – review & editing:** Peizhen Yang, Ericka Havecker, Matthew Bauer, Bill Hendrix, Jill Deikman.

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
