## [Decision Letter · Decision Letter 0]

25 Mar 2021

PONE-D-20-40091

Beyond identity: understanding the contribution of the 5’ nucleotide of the antisense strand to RNAi activity

PLOS ONE

Dear Dr. Yang,

Thank you for submitting your manuscript to PLOS ONE. After careful consideration, we feel that it has merit but does not fully meet PLOS ONE’s publication criteria as it currently stands. Therefore, we invite you to submit a revised version of the manuscript that addresses the points raised during the review process.

As you can see from their comments listed below, both reviewers indicated that your manuscript is highly interesting but also mentioned a few concerns that need to be properly addressed before the manuscript can be accepted for publication.

We look forward to receiving your revised manuscript.

Kind regards,

Sebastien Pfeffer, PhD

Academic Editor

PLOS ONE

Journal Requirements:

1. Please ensure that your manuscript meets PLOS ONE's style requirements, including those for file naming. The PLOS ONE style templates can be found athttps://journals.plos.org/plosone/s/file?id=wjVg/PLOSOne_formatting_sample_main_body.pdf and

'The authors have declared that no competing interests exist.'

We note that one or more of the authors are employed by a commercial company: Bayer Crop Science.

Additional Editor Comments (if provided):

Reviewers' comments:

Reviewer's Responses to Questions

**Comments to the Author**

1. Is the manuscript technically sound, and do the data support the conclusions?

Reviewer #1: Yes

Reviewer #2: Yes

2. Has the statistical analysis been performed appropriately and rigorously? 

Reviewer #1: No

Reviewer #2: Yes

3. Have the authors made all data underlying the findings in their manuscript fully available?

Reviewer #1: No

Reviewer #2: Yes

4. Is the manuscript presented in an intelligible fashion and written in standard English?

Reviewer #1: Yes

Reviewer #2: Yes

5. Review Comments to the Author

Reviewer #1: In this manuscript, P. Yang and colleagues explore the role of nucleotide identity at the 5´ end of the guide strand of an siRNA in tobacco. With the development of a novel technique for experimental administration of synthetic siRNAs in leaves (deposition of the siRNA solution on the leaf, followed by sandpaper abrasion), they can deliver fairly consistent amounts of siRNAs, which allows them to compare the efficiency of siRNA sequence variants.

The manuscript is clearly written, and it addresses an important question (the effect of 5´ nucleotide identify had been measured by several studies in animals, but not in plants). The results are convincing, and they raise novel, interesting questions. That manuscript thus deserves to be published in PLoS One, once a few issues have been corrected (listed below).

Major issues:

1. Line 116 (also on line 290): does the calculation really only consider the stacking energy? And not the pairing energy between bases on the two strands? The asymmetry rule (as has been defined in animal systems) relies on differential pairing energy (which depends on the stacking energy, but also, obviously, on the hydrogen bonds between the bases). This is an important point, because the calculated asymmetries in this manuscript may be erroneous if they only rely on the stacking term: please clarify.

2. Line 271: a very interesting finding (the G-C pair behaves very differently from the C-G pair). This result should be discussed in detail, and the manuscript would benefit from a tentative molecular interpretation of this surprising fact.

3. Lines 375 and 376: this information can be easily extracted from miRBase (http://www.mirbase.org/), at least for N. tabacum. Please provide a clear assessment of the existence or not of such miRNAs in N. tabacum (with the limitation that this will just be a proxy for N. benthamiana), rather than a declaration that the authors are "unaware of [their existence]", which is way too vague.

4. Statistical assessment of the differences shown in histograms (Fig. 1B-D; Fig. 3B; Fig. 4B) should be provided (this also applies to Fig. 2B, but apparently the authors have done the analysis - just, I don't understand the way the results are displayed: see minor issue #16). All these datasets can probably be analyzed with an ANOVA (two-way ANOVA for Fig. 3B and 4B; one-way ANOVA for the other ones), providing that applicability conditions are met.

Minor issues:

1. Line 44: siRNAs do not "associate" with the RISC complex: they are part of it.

2. Line 49: siRNAs do not always have a 3´ hydroxyl (in plants where it has been studied, they are 2´-O-methylated by Hen1 after RISC loading).

3. Line 49: the AGO proteins are not thought to sort duplexes, and select one strand as the guide. All these activities are actually thought to be performed by the AGO loading machinery (in particular, the machinery orients the duplex in such a way that one strand will be nested in the groove of the AGO protein, where it will remain stably bound: that's the guide strand; the loading step thus determines which strand will be the guide and which strand will be the passenger). Similarly on line 86: strand selection is not a matter of "retention", but rather: a matter of loading geometry.

4. Line 80: the expression "the less stable 5´ end" is a bit misdleading. Strictly speaking, it means that the 5´ end of that strand is biochemically less stable (i.e.: it would be degraded, for example by nucleases). What the authors mean is that this strand's 5´ end is "less stably paired", which is not the same thing.

5. Similarly on line 82: it is necessary to explain that the analyzed score here is the differential pairing free enegy (rather than the differential free energy).

6. Line 94: I do not understand what the authors mean with "in the context of thermodynamic stability".

7. Line 97: "more similar than previously thought". This is not clear to me (I am not aware of people claiming that the asymmetry rule does not apply to plant siRNAs). At the very least, the authors should provide a bibliographical reference to illustrate that (disproven) claim.

8. Line 108 (also on line 339): I don't know what the "Reynolds score" is (a description is needed here, in addition to the bibliographical references).

9. Lines 113-114: the URL is not valid.

10. Line 121: Light intensity is not measured in micromoles (!).

11. Line 135: please specify what "protease inhibitor" has been used (there are many of them).

12. Line 152: the amount of RNA used for reverse-transcription should be specified (explaining that the RNA solution concentration is lower than 200 ng/μL is much less useful).

13. Line 159: the amplification efficiency in a qPCR reaction is never equal to 2, but that is what has apparently been used for this calculation. Amplification efficiency can be estimated from the qPCR data itself, providing better ways of calculating cDNA quantities: please use a better method.

14. Line 211: a probable typo ("AAG" should apparently read "AAC").

15. Line 247 (also on line 362): please avoid the term "slicer mutant" ("Slicer" is usually the name given to the catalytic AGO protein, not to its RNA guide).

16. Line 284 (last sentence of the legend for Fig. 2): I do not understand the labels. Please clarify.

17. Line 351 (last sentence of the legend for Fig. 5A): this is really unclear. It would be much better to simply draw the mutated duplexes, so the reader can easily understand what the mutation looks like (also, note that "U-A" is not a mismatch).

18. Lines 365 and 366: please provide an AGO gene count for N. benthamiana too (not just Arabidopsis and rice, which are not being studied in this manuscript).

19. Line 401: what is a "miRNA designer"?

Reviewer #2: In this manuscript, Yang and colleagues investigated whether 5’-nt identity is an important determinant of posttranscriptional gene silencing via topically-applied siRNAs in plants. They used sandpaper abrasion to introduce various synthetic, 21 bp siRNA duplexes to N. benthamiana leaves, then evaluated the silencing of marker genes homologous to those siRNAs. Deficiency for an endogenous target in N. benthamiana, CHLH, produces small white spots due to loss of chlorophyll. Silencing of a GFP transgene, on the other hand, reduces GFP fluorescence in N. benthamiana leaves, leading to red fluorescent spots under UV illumination (from chlorophyll). The authors measured the molecular effects of gene silencing using GFP mRNA RT-qPCR and GFP protein detection via western blot.

Early studies in this field found that synthetic siRNAs directly trigger silencing when introduced into plants via biolistic delivery (Klahre et al. 2002 PNAS). The authors of the current manuscript have optimised an ingenious sandpaper abrasion method to deliver synthetic siRNAs into plant leaves and to trigger silencing. The authors present rigorous findings on how siRNA duplex thermostability influences the efficiency of RNA silencing by these exogenous siRNAs. Based on these findings, engineers might design better siRNAs by lowering the 5’ stability of the antisense end to activate siRNA duplexes, promoting incorporation of the correct strand into the RISC complex that specifically targets mRNAs during RNA silencing. Understanding this process has important implications for crop protection and weed control, as cited by the authors.

This manuscript will be of interest to the field of RNA silencing in plants, particularly to researchers looking for advice for applications involving siRNA delivery.

Minor issues:

1) The authors should cite Klahre et al. 2002 PNAS (10.1073/pnas.182204199), one of the first studies to show that synthetic siRNA duplexes can directly target genes for silencing in N. benthamiana. Manavella et al. 2012 PNAS (10.1073/pnas.1200169109) looked at the question of miRNA/miRNA* duplex asymmetry in plants, which is also relevant to the authors' manuscript.

2) Fig. 1A shows rudimentary schematics of the targets GFP and NbCHLH. The authors cite Papenbrock et al. (2000) Plant physiol. for reference to NbCHLH silencing phenotypes observed in plants deficient for this magnesium chelatase subunit.

a. Please provide a more precise diagram of the NbCHLH gene and predicted mRNA transcripts (exon-intron structure, start codon, stop codon, predicted UTRs, alternative splice forms). Presumably both the GFP transgene and NbCHLH endogenous locus have promoter regions, which could be included in the diagram.

b. For clarity concerning the GFP targeting strategy (oligo sequences) and silencing effects, the authors might consider including the data in Fig. 2 as subpanels of Fig. 1, where GFP fluorescence phenotypes are shown. In either case, it would be helpful to see representative images for the GFP silencing experiments described in Fig. 2, alongside the GFP silenced area quantification.

c. Since 2012 there is an N. benthamiana draft genome. I recommend that the authors reference NbCHLH by its gene identifier at:

https://solgenomics.net/organism/Nicotiana_benthamiana/genome in addition to the Nbv6.1trP75862 transcript identifier from http://sefapps02.qut.edu.au/ cited on page 12.

d. Here and elsewhere (Figs 1A, 2A, 3A, 5A), I recommend that the authors show antisense strands of siRNA duplexes right-side up, rather than with the nucleotide symbols inverted. To indicate nucleic acid orientation, it would be better to add 5'-to-3' and 3'-to-5' annotations, as appropriate, at siRNA termini following the style of most journals.

3) Fig. 1C is labeled with "Blue light" for GFP fluorescence viewing. The authors must mean "Ultraviolet" illumination.

4) Fig. 3 there appears to be an error on the y-axis label. "CheH Silenced" should be "CHLH silenced".

5) Fig. 4A shows a diagram of GFP:CHLH gene silencing in protoplasts. The illustration is not clear in this form. The key features of the procedure should be labeled with specific text. Using the same arrow type to indicate siRNA delivery and reduced GFP expression is confusing. Do the authors have any images of the real protoplasts tested during these experiments?

6) Fig. 5A, like above in Fig. 1A, please use a more precise diagram to depict NbCHLH.

6. PLOS authors have the option to publish the peer review history of their article (what does this mean?). If published, this will include your full peer review and any attached files.

Reviewer #1: No

Reviewer #2: No

---

## [Author Response · Author response to Decision Letter 0]

9 Jun 2021

The authors would like to thank both reviewers for their time reviewing the manuscript and providing valuable feedback to help us improve the manuscript. 

 Please see the detailed response in the " responses to the reviewers.docx".

---

## [Decision Letter · Decision Letter 1]

7 Jul 2021

PONE-D-20-40091R1

Beyond identity: understanding the contribution of the 5’ nucleotide of the antisense strand to RNAi activity

PLOS ONE

Dear Dr. Yang,

Thank you for submitting your manuscript to PLOS ONE. After careful consideration, we feel that it has merit but does not fully meet PLOS ONE’s publication criteria as it currently stands. Therefore, we invite you to submit a revised version of the manuscript that addresses the points raised during the review process.

As you will see, Reviewer 1 still has concerns that mostly pertain to semantic or graphic representation problems. You need to take those into account before the manuscript can be formally accepted.

We look forward to receiving your revised manuscript.

Kind regards,

Sebastien Pfeffer, PhD

Academic Editor

PLOS ONE

Journal Requirements:

Reviewers' comments:

Reviewer's Responses to Questions

**Comments to the Author**

1. If the authors have adequately addressed your comments raised in a previous round of review and you feel that this manuscript is now acceptable for publication, you may indicate that here to bypass the “Comments to the Author” section, enter your conflict of interest statement in the “Confidential to Editor” section, and submit your "Accept" recommendation.

Reviewer #1: (No Response)

Reviewer #2: All comments have been addressed

2. Is the manuscript technically sound, and do the data support the conclusions?

Reviewer #1: Yes

Reviewer #2: Yes

3. Has the statistical analysis been performed appropriately and rigorously? 

Reviewer #1: No

Reviewer #2: Yes

4. Have the authors made all data underlying the findings in their manuscript fully available?

Reviewer #1: Yes

Reviewer #2: Yes

5. Is the manuscript presented in an intelligible fashion and written in standard English?

Reviewer #1: No

Reviewer #2: Yes

6. Review Comments to the Author

Reviewer #1: The authors have now corrected several of the points I had raised in my initial review, but some still remain. Some of them really make the figures un-understandable (the usage of Student's t-test to "separate" (?) means; the undocumented usage of one-letter or two-letter codes to denote statistical significance). These points really need to be improved for the manuscript to be understandable.

Detailed remarks:

1. The new version of the manuscript now says that plant siRNAs bear 2´-O-methylation on their 3´ ends, which is good (lines 48-50), but the new formulation seems to imply that animal siRNAs always have a 3´ hydroxyl, which is not true at least in Drosophila. Please clarify.

2. The initial version of the manuscript was quantifying light intensity in micromoles. I suspected it was a mistake and asked for correction, and the new version now measures light intensity in "µmol m-2 s-1". Did the authors really get my point? Moles (and micromoles) measure amounts of particules (e.g., one mole of water is 6.02e23 molecules of water). Light intensity is measured in candelas, and light power is measured in watts. Not moles (or "moles per square meter and per second"). Unless the authors are counting moles of photons? Please clarify.

3. The formulation "means were separated using Student's t-test" (lines 188, 190, 245, 304, 320 and 358) is extremely obscure. What did the authors exactly do? Student's t-test, just like the ANOVA, can assess whether the observed difference is significant (meaning that the sampled ideal populations have different means), except that Student's t-test can only compare 1 group to 1 group (one-way ANOVA can compare several groups to each other). So what has been "separated" here? Why has the Student's t-test been applied after the ANOVA?

4. The idea of indicating significant differences with a letter is a bit strange, it really doesn't help understanding. Why not simply adding asterisks on the bars with p-value < 0.05, as is classically done? The description of this counter-intuitive lettered code is obfuscated, making it hard for the reader to understand what is going on (see for example in the legend for Figure 1, line 246: "Means not followed by the same letter indicate statistical significance", but the figure does not indicate any mean value followed by any letter; I can only see letters "A" and "B" above the bars of the barplot, without any mean being written out). Figure 2B adds one more unexplained level of complexity, introducing 2-letter codes ("AB" and "BC") instead of 1-letter codes, and I cannot make sense out of it. The concept of displaying significance is quite simple, so please don't make it uselessly complicated with this undocumented letter code.

5. In response to my initial comments, the authors explained that the compared free energies at the duplex' extremities are not just stacking energies, but also base-pairing energies (which makes a lot of sense of course). But then why does the updated text still read (on lines 310-312) that the comparison was about "stacking energies of the first 5 pairs" and an obscure "free end loop"? What is this loop? Please be more rigorous on the terms (I thought I was clear in my first evaluation): stacking energy is just one term in the total pairing energy, and most likely the authors are comparing the total pairing energies here. Also, please detail what the "free en loop" is, and what is its role in this energy comparison (I really have no clue what this loop thing is).

Reviewer #2: In this revised manuscript, Yang and colleagues investigated whether 5’-nt identity is an important determinant of posttranscriptional gene silencing triggered by topically-applied siRNAs in plants. They used sandpaper abrasion to introduce synthetic 21 bp siRNA duplexes to N. benthamiana leaves, then evaluated silencing of genes homologous to those siRNAs. Deficiency for an endogenous target in N. benthamiana, CHLH, produces white spots. Silencing of a GFP transgene, on the other hand, reduces GFP fluorescence in N. benthamiana leaves, leading to red fluorescent spots under blue light illumination (concerning this revision, I thank the authors for correcting my mistake about the GFP5 excitation wavelength). The authors measured this gene silencing by detecting GFP mRNA levels (RT-qPCR) and GFP protein (western blot).

In their revised manuscript the authors addressed all the minor issues that I raised, and several substantive concerns of the other referee. They now cite the missing studies, have better annotated figures and have explained the other technical aspects that were confusing in the original manuscript.

The authors present rigorous findings on how siRNA duplex thermostability influences the efficiency of RNA silencing by exogenous siRNAs. Understanding this process has implications for crop protection and weed control. This manuscript will be of interest to the field of RNA silencing in plants, particularly to researchers looking for advice on applications involving siRNA delivery.

7. PLOS authors have the option to publish the peer review history of their article (what does this mean?). If published, this will include your full peer review and any attached files.

Reviewer #1: No

Reviewer #2: No

---

## [Author Response · Author response to Decision Letter 1]

16 Aug 2021

Dear Dr. Sebastien Pfeffer, 

Thank you for your leading this review process of our submission “Beyond identity: understanding the contribution of the 5’ nucleotide of the antisense strand to RNAi activity” to Plos One journal. We would like to sincerely thank both reviewers for their valuable time going through the revised manuscript and providing additional feedbacks. We appreciate reviewer 2’s recommendation of the revised version to the journal for publication, as well as reviewer 1’s acceptance of updates and the additional comments. Please see below our responses to reviewer 1’s comments (including the statistical calculations and terminal free energy, etc) and the corresponding changes made in the manuscript. 

PONE-D-20-40091R1

Beyond identity: understanding the contribution of the 5’ nucleotide of the antisense strand to RNAi activity

Reviewer #1: The authors have now corrected several of the points I had raised in my initial review, but some still remain. Some of them really make the figures un-understandable (the usage of Student's t-test to "separate" (?) means; the undocumented usage of one-letter or two-letter codes to denote statistical significance). These points really need to be improved for the manuscript to be understandable.

Detailed remarks:

1. The new version of the manuscript now says that plant siRNAs bear 2´-O-methylation on their 3´ ends, which is good (lines 48-50), but the new formulation seems to imply that animal siRNAs always have a 3´ hydroxyl, which is not true at least in Drosophila. Please clarify.

The authors thank the reviewer for catching this. We updated the sentence to “siRNA duplexes are generally 20-25 nt in length, have 2nt 3’ overhangs and are bounded by 5’-phospate and 3’ hydroxyl moieties and additionally have 2’-O-methylation at their 3’ ends” to clarify the point that the siRNAs regardless of kingdom have 3’ hydroxyl and 2’ O- methylation by removing the “plant and animal” aspect of the original sentence.

2. The initial version of the manuscript was quantifying light intensity in micromoles. I suspected it was a mistake and asked for correction, and the new version now measures light intensity in "µmol m-2 s-1". Did the authors really get my point? Moles (and micromoles) measure amounts of particules (e.g., one mole of water is 6.02e23 molecules of water). Light intensity is measured in candelas, and light power is measured in watts. Not moles (or "moles per square meter and per second"). Unless the authors are counting moles of photons? Please clarify.

We did add the word “intensity” into our methods to clarify. The standard units for greenhouse lighting is measured in µmol m-2 s-1 or µmol/m2/s and for most if not all plant journals. We appreciate the attention to detail and can offer this explanation. The light intensity measurements are different for visible light for humans as opposed to greenhouse lighting. For human visible light, when describing light quantity or light intensity units - foot candles, lux , lumens etc are used. For greenhouse lighting, light intensity for plants is measured in a different range (400-700nm) and is referred to as Photosynthetic Active Radiation (PAR). The unit for measuring the PAR or instantaneous light incident upon a surface is micromoles per square meter per second (or µmol/m2/s) - this is the amount of energy (photons or particles of light) hitting a square meter every second. One source that explained this well was http://www.greenhouse.cornell.edu/structures/factsheets/Greenhouse%20Lighting.pdf. A scientifically published book chapter explaining PAR and a conversion table is https://www.sciencedirect.com/topics/agricultural-and-biological-sciences/photosynthetically-active-radiation. In plant science, researchers programming a greenhouse or growth chamber to recreate this study would very likely use a µmol/m2/s value to program the chamber or greenhouse settings. 

3. The formulation "means were separated using Student's t-test" (lines 188, 190, 245, 304, 320 and 358) is extremely obscure. What did the authors exactly do? Student's t-test, just like the ANOVA, can assess whether the observed difference is significant (meaning that the sampled ideal populations have different means), except that Student's t-test can only compare 1 group to 1 group (one-way ANOVA can compare several groups to each other). So what has been "separated" here? Why has the Student's t-test been applied after the ANOVA?

There may be a misunderstanding of language around statistical analysis used in these studies. The reviewer is correct that one-way ANOVA compares means of several groups using the F-test statistic. A significant F-test indicates significant difference among all groups, a global test of significance, but does not provide specific test of the difference between any two groups. We evaluated the F-test and if the P-value was < 0.05, we then performed statistical tests of all possible pairwise comparisons using a t-test-. This method commonly referred to as Fischer’s-protected least significant difference. This is a classical method that provides protection for multiple testing and has been employed in across scientific disciplines for more than 40 years. This algorithm in implemented in the statistical software package we utilized (JMP12, SAS institute). We believe this method is appropriately used in these studies. We have altered the language in the text at each line pointed out by the reviewer to clarify this procedure.

4. The idea of indicating significant differences with a letter is a bit strange, it really doesn't help understanding. Why not simply adding asterisks on the bars with p-value < 0.05, as is classically done? The description of this counter-intuitive lettered code is obfuscated, making it hard for the reader to understand what is going on (see for example in the legend for Figure 1, line 246: "Means not followed by the same letter indicate statistical significance", but the figure does not indicate any mean value followed by any letter; I can only see letters "A" and "B" above the bars of the barplot, without any mean being written out). Figure 2B adds one more unexplained level of complexity, introducing 2-letter codes ("AB" and "BC") instead of 1-letter codes, and I cannot make sense out of it. The concept of displaying significance is quite simple, so please don't make it uselessly complicated with this undocumented letter code.

The use of letter codes to indicate significance is well-documented and commonly used in plant biology and many other disciplines. In the latest issue of PLOS One, Potcho et al (2021) use letter codes and Fishers protected LSD in the same way we utilized this common method on our studies. We revised the letter codes to lower case to meet PLOS One requirements, but we believe this statistical procedure is sound and supported by years of use in the literature, in statistical textbooks, and in the statistical software package used in this study (JMP12, SAS institute).

Potcho PM, Okpala NE, Korohou T, Imran M, Kamara N, et al. (2021) Nitrogen sources affected the biosynthesis of 2-acetyl-1-pyrroline, cooked rice elongation and amylose content in rice. PLOS ONE 16(7): e0254182. https://doi.org/10.1371/journal.pone.0254182

5. In response to my initial comments, the authors explained that the compared free energies at the duplex' extremities are not just stacking energies, but also base-pairing energies (which makes a lot of sense of course). But then why does the updated text still read (on lines 310-312) that the comparison was about "stacking energies of the first 5 pairs" and an obscure "free end loop"? What is this loop? Please be more rigorous on the terms (I thought I was clear in my first evaluation): stacking energy is just one term in the total pairing energy, and most likely the authors are comparing the total pairing energies here. Also, please detail what the "free en loop" is, and what is its role in this energy comparison (I really have no clue what this loop thing is).

The authors appreciate the feedback from the reviewer being more rigorous on the description of the “total pairing energy”. Since the calculation of the delta G has been described in the earlier sentence in the same paragraph, we have updated the sentence of “as the differential free energy between two ends of the duplex of the first 5 pairs of the sequence and the free end loops. Ct = 10uM, Na+ at 1M, and @ 37C.” to “as the difference in thermodynamic stabilities at each end”. 

Regarding the terminal stability (dG, kcal.mole-1) at the each end of the siRNA duplex, as described in the materials and methods section, we used the software of DINA melt (http://www.unafold.org/Dinamelt/applications/two-state-melting-hybridization.php) by summing the first four nearest-neighbor energies and the free energy of 2nt overhang. The free end loops refer to the 3’ 2nt overhangs that affect the free energy of the end of RNA structure. In many studies comparing siRNA properties with large dataset, typically the last 2nt at 3’ end of both strands are set at “TT” so to cancel the energy contribution from the overhangs and simplify the calculation of ddG to just the 4 nearest neighbor energies from the stem region. In this study, we didn’t use the generic “TT” as the overhang but rather the actual matching sequences from the genes, so the terminal energy include the contribution of 2’ nt overhangs in addition to the four nearest-neighbor energies at the 5’ end of the siRNA. 

In original publication by Turner and Matthews (see below) on the nearest-neighbour model, the overhang shown as below is described as an exterior loop (as is highlighted in the red circle). We borrowed this concept of “free-end loop“ to describe the overhang. We do agree with the reviewer that this description seems confusion, therefore we have updated in the text as overhang, to make the description clear.

Turner, D. H., & Mathews, D. H. (2010). NNDB: the nearest neighbor parameter database for predicting stability of nucleic acid secondary structure. Nucleic acids research, 38(Database issue), D280–D282. https://doi.org/10.1093/nar/gkp892

We look forward to hearing from you regarding our re-submission. We would be glad to respond to any further questions and comments that you may have. Please address all correspondence concerning this manuscript to me at peizhen.yang@bayer.com. 

Sincerely,

Peizhen Yang, Ph.D.

Applied Cell Biology Team Lead

608-695-6167

Peizhen.yang@bayer.com

---

## [Editor Report · Decision Letter 2]

18 Aug 2021

Beyond identity: understanding the contribution of the 5’ nucleotide of the antisense strand to RNAi activity

PONE-D-20-40091R2

Dear Dr. Yang,

We’re pleased to inform you that your manuscript has been judged scientifically suitable for publication and will be formally accepted for publication once it meets all outstanding technical requirements.

Kind regards,

Sebastien Pfeffer, PhD

Academic Editor

PLOS ONE

---

## [Editor Report · Acceptance letter]

27 Aug 2021

PONE-D-20-40091R2 

Beyond identity: understanding the contribution of the 5’ nucleotide of the antisense strand to RNAi activity 

Dear Dr. Yang:

I'm pleased to inform you that your manuscript has been deemed suitable for publication in PLOS ONE. Congratulations! Your manuscript is now with our production department. 

Kind regards, 

on behalf of

Dr. Sebastien Pfeffer 

Academic Editor

PLOS ONE